# Into a Deeper Understanding of CYP2D6’s Role in Risperidone Monotherapy and the Potential Side Effects in Schizophrenia Spectrum Disorders

**DOI:** 10.3390/ijms25126350

**Published:** 2024-06-08

**Authors:** Mariana Bondrescu, Liana Dehelean, Simona Farcas, Patricia Alexandra Dragan, Carla Andreea Podaru, Laura Popa, Nicoleta Andreescu

**Affiliations:** 1Department of Neurosciences-Psychiatry, “Victor Babes” University of Medicine and Pharmacy, Eftimie Murgu Square 2, 300041 Timisoara, Romania; mariana.bondrescu@umft.ro; 2Timis County Emergency Clinical Hospital “Pius Brinzeu”, Liviu Rebreanu 156, 300723 Timisoara, Romania; patricia.dragan7@gmail.com; 3Doctoral School, “Victor Babes” University of Medicine and Pharmacy, Eftimie Murgu Square 2, 300041 Timisoara, Romania; laura.popa@umft.ro; 4Discipline of Medical Genetics, Department of Microscopic Morphology, Center of Genomic Medicine “Victor Babes” University of Medicine and Pharmacy, Eftimie Murgu Square 2, 300041 Timisoara, Romania; farcas.simona@umft.ro (S.F.); andreescu.nicoleta@umft.ro (N.A.); 5Independent Researcher, 300523 Timiosara, Romania; carla.podaru@gmail.com

**Keywords:** schizophrenia spectrum disorders, CYP2D6, risperidone treatment

## Abstract

Schizophrenia spectrum disorders (SSD) are a group of diseases characterized by one or more abnormal features in perception, thought processing and behavior. Patients suffering from SSD are at risk of developing life-threatening complications. Pharmacogenetic studies have shown promising results on personalized treatment of psychosis. In the current study, 103 patients diagnosed with SSD treated with risperidone as antipsychotic monotherapy were enrolled. Socio-demographics and clinical data were recorded, and laboratory tests and genotyping standard procedure for cytochrome P450 (CYP) 2D6*4 were performed. Patients were evaluated by the Positive and Negative Syndrome Scale (PANSS) on admission and at discharge. Based on the reduction in the PANSS total score, subjects were divided into non-responders, partial responders and full responders. Only 11 subjects had a full response to risperidone (10.67%), 53 subjects (51.45%) had a partial response, and 39 participants (37.86%) were non-responders. Patients at first episode psychosis showed significantly higher levels of blood glucose and prolactin levels, while chronic patients showed significantly higher LDL levels. Adverse drug reactions (ADR) such as tremor and stiffness significantly correlated with genetic phenotypes (*p* = 0.0145). While CYP2D6 showed no impact on treatment response, ADR were significantly more frequent among poor and intermediate metabolizers.

## 1. Introduction

Schizophrenia spectrum disorders (SSD) are a group of diseases characterized by one or more abnormal features in the following domains: perception (hallucinations or illusions), thought processing (delusions, disorganized thinking, negative symptoms) and behavior (motor behavior disturbances or disorganized behavior) that are in line with distortions in the process of thinking [1].

Epidemiological data on the prevalence of different psychotic disorders are scant [2], with most of the studies focusing on schizophrenia as a representative of the spectrum. Thereby, schizophrenia is around 1% prevalent, with regional variations [3] with a relative heterogenous incidence between different countries, within cities and into populations as reported in a study from the United States of America [4]. Slightly different results have been shown in another study done over a period of 8 years in Spain, showing a prevalence of 6.2 in 1000 persons. Interestingly, schizophrenia was 76% more common among men than women, with an incidence two times higher in men than women [5]. In 2019, 23.6 million people have been diagnosed with schizophrenia, which implies an increase of 65.85% from 1990 globally [6]. Therefore, significant healthcare costs and economic burden relate to schizophrenia [7].

Moreover, persons diagnosed with schizophrenia are at moderate to high risk of developing severe somatic diseases, such as diabetes mellitus, cancer, cerebrovascular and ischemic heart diseases soon after the psychosis onset [8]. Not only do people suffering from schizophrenia experience tremendous distress and impairment in the majority of life domains, but they are also at a 2 to 3 times higher risk of premature death than the general population according to World Health Organization (WHO) [9]. The intricate presentation of schizophrenia is consistent with its complex pathogenesis, influenced mainly by genetic, biological and environmental factors [10].

The contribution of genetic factors in the treatment efficacy of psychosis has been extensively studied. Together with age of onset, illness duration, disease severity, level of education, smoking habits, drug abuse and diet, genetic factors play a significant role in the interindividual response to antipsychotic treatment. Several second-generation antipsychotics have been frequently used in the treatment of schizophrenia spectrum disorders, among which risperidone exhibits a strong antagonist effect for both D2 and 5-HT2 receptors, consequently with important therapeutic effects on the positive and negative symptoms of schizophrenia [11]. In addition, risperidone’s α1, α2 and H1 antagonism are responsible not only for anxiolytic and hypnotic actions, but also for some of the unpleasant adverse reactions such as orthostatic hypotension and sedation [12]. Risperidone has proven superior to first-generation antipsychotics in controlling schizophrenic symptoms in general and in amphetamines-associated psychosis [13,14]. In comparison to clozapine, it has proven better efficacy on depressive symptoms in chronic schizophrenia [15]. Not only has risperidone shown efficacy in chronic psychosis, but also in first-episode psychotic patients [16]. Additionally, risperidone has proven a good profile of tolerability in the long-term treatment of schizophrenia [17] with lower risk of severe side effects than conventional antipsychotics in general [18]. In contrast, when compared with haloperidol, patients treated with risperidone exhibited higher levels of total cholesterol and experienced more significant weight gain than those receiving haloperidol [18,19]. Another considerable risk in the treatment with risperidone is the possibility of extrapyramidal side effects (EPS), such as dystonia, parkinsonism, akathisia, and tardive dyskinesia. Although the appearance of EPS may be a sign of therapeutic levels’ achievement [20], they decrease the quality of life [21], and seem to occur more often in patients treated with risperidone than other second-generation antipsychotics [22], but less than in those treated with the conventional ones [23]. Several studies have tried to shed light on the different aspects of risperidone treatment in schizophrenia spectrum disorders, stressing the role of pharmacogenetics in obtaining better outcomes and a more personalized approach [24,25].

Pharmacogenetic studies focus on how treatment response is altered according to the gene’s variability involved in drug transformations and availability [26,27]. A considerable number of drugs, including antipsychotics, are metabolized by P450 hepatic cytochrome (CYP) enzymes. These enzymes are extremely complex and have more than 100 allelic variants for CYP2D6 [27], which is the main metabolization pathway for risperidone, along with CYP3A4 and CYP3A5, but to a smaller extent [28]. Following 9-hydroxylation by CYP2D6, risperidone is converted into 9-hydroxyrisperidone, also known as paliperidone, and used in racemic mixtures as a sole antipsychotic in schizophrenia treatment. Although 9-hydroxyrisperidone is its primary active metabolite, exerting the most important therapeutic action [29,30], another metabolite of risperidone is 7-hydroxyriperidone, with lesser metabolic activity. Following the metabolization of risperidone to 9-hydroxyrisperidone, a chiral center is introduced that results in enantiomer formation. Consequently, in vivo studies have shown that the dominant role of CYP2D6 is in the formation of (+)-9-hydroxyrisperidone, while the dominant role of CYP3A4 is in the formation of (−)-9-hydroxyrisperidone [31]. Apart from hydroxylation, another metabolic pathway for risperidone is oxidative N-dealkylation, which results in inactive metabolite formation [32]. Thus, the active moiety of risperidone is 9-hydroxyrisperidone with several implications in treatment response and the risk for adverse effects. A study on 82 patients treated with risperidone showed a significant correlation between active moiety plasma levels and risperidone dosage. Higher plasma levels seemed to also be important predictors for the incidence of extrapyramidal side effects [33]. Although the efficacy of risperidone can be influenced by several factors, CYP2D6 polymorphisms seem to play an important role [34,35]. While persons with two null alleles appear to have no enzymatic activity (poor metabolizers-PM), intermediate metabolizers (IM) with either two deficient alleles or one gene with absent enzymatic activity and one with normal activity (EM) may have partial enzymatic activity, meaning a relatively lower risk for adverse drug reaction (ADR) than the previous category [36]. According to data in the literature, several polymorphisms showed importance in treatment outcomes to risperidone, with CYP2D6*3, *4, *10 and *14 being the most frequently encountered as intermediate and poor phenotypes, whereas CYP2D6*2 and *5 were relatively rarely encountered in psychotic patients [33,37]. Regarding CYP2D6 variability across population, the CYP2D6*4 variant was frequently found in the European population [38], while the *5 variant was particularly encountered in the population of South Africa [39,40], *10 in East Asian people, and *17 and *29 in the Black African population, while the *41 variant was frequent among both European individuals and western and southern Asian population [38,41]. A study by Riedel et al. found that CYP2D6*4, *6 and *14 polymorphisms did not affect the clinical effects of risperidone but influenced risperidone’s active moiety levels [33]. In addition, the same impact on 9-hydroxyrisperidone and risperidone plasma levels was observed in homozygote patients with CYP2D6*10 [37]. As a result, intermediate metabolizers and poor metabolizers will experience longer exposure to risperidone’s active moiety levels with a consequent accumulation of the drug, increasing the risk for adverse effects. The most important drug reactions to risperidone, frequently associated with CYP2D6 polymorphisms, were extrapyramidal side effects, hyperprolactinemia and metabolic syndrome. Moreover, patients with parkinsonism and dystonia had significantly higher plasma levels of 9-hydroxyrisperidone [32]. While the expression of CYP2D6*4 was associated with tardive dyskinesia [42], hyperprolactinemia was not clearly associated with a certain variant, but instead with various CYP2D6 polymorphisms, and was more often representative in pediatric patients treated with risperidone [43,44]. However, hyperprolactinemia seems to be significantly increased in patients receiving risperidone after 8 weeks of treatment, according to a study from 2016 [45]. Thus, previous studies have tried to explain the role of CYP2D6*4 in the response to risperidone, but most of the results are inconsistent [46,47], suggesting that further investigation is required.

Considering that CYP2D6*4 is one of the most frequent variants in European population [38,41] and also responsible for 70–90% of all the nonfunctional phenotypes [43], the current study hypothesized that therapeutic response to risperidone is influenced by CYP2D6*4 polymorphisms, and that PM and IM individuals experience ADR more frequently. Therefore, the main study outcome is the evaluation of risperidone treatment response by PANSS, based on the genetic polymorphisms of CYP2D6*4.

## 2. Results

### 2.1. Demographics and CYP2D6 Genotyping Characteristics

A total number of 107 patients were recruited for the current study. Following primary evaluation, one patient was excluded due to incomplete protocol evaluation, two dropped out of the study because of non-compliance, and one was withdrawn by reason of unclear diagnosis. Consequently, 103 subjects remained in the study and underwent the data analysis. The participants received risperidone as antipsychotic monotherapy. Figure 1 represents subjects’ distribution according to phenotypes.

The subjects were divided into three categories based on the treatment response evaluated according to the reduction in the total score of PANSS from admission to discharge (duration of hospitalization was between 2 weeks and 10 weeks). Only 11 subjects had a full response to risperidone treatment (10.67%), 53 subjects (51.45%) had a partial response, and 39 participants (37.86%) were qualified as non-responders. The socio-demographics of the participants included in the study are presented in Table 1. There were no statistically significant differences between the ages of the participants.

With respect to the body mass index (BMI) measure after 2 weeks of treatment, similar means were obtained between the three groups: 25.63 (SD ± 5.19), 26.16 (SD ± 7.08) and 23.61 (SD ± 5.05) corresponding to non-responders, partial responders and full responders, respectively. A statistical analysis showed no significant differences in age of onset among the three groups (*p* = 0.646).

The distribution of diagnoses, which included schizophrenia, schizoaffective disorder, schizophreniform disorder and delusional disorder, did not significantly vary across the response groups (*p* = 0.304). The number of episodes, categorized as 1, 2–3, and more than 3, also showed no significant association with the level of response (*p* = 0.547). The clinical characteristics of the study population are listed in Table 2.

The frequency of ADR, which were categorized as none, tremor, stiffness, and both tremor and stiffness, did not differ significantly across response categories (*p* = 0.636), although there was a noticeable but non-significant trend towards more frequent tremor-related reactions among full responders. Likewise, risperidone dosage (≤2 mg/day, 3–4 mg/day, ≥5 mg/day) was not significantly different across the groups (*p* = 0.795).

Full responders needed fewer days to establish significant reduction in PANSS total scores, with 54.55% having a hospitalization duration of ≤14 days, compared to 28.20% of non-responders and 13.21% of partial responders. Conversely, longer hospitalizations (15–28 days and >42 days) were more common in partial responders. However, after Bonferroni correction, the differences observed in the duration of hospitalization did not reach significance.

Statistical analyses of biochemical markers determined after two weeks of risperidone treatment revealed significant differences in blood glucose and triglycerides across the groups, as indicated by the ANOVA results, which are presented in Table 3. Blood glucose levels differed significantly (*p* = 0.02), with full responders having the lowest mean blood glucose level at 74.82 mg/dL, compared to 76.96 mg/dL in partial responders and 78.13 mg/dL in non-responders.

Similarly, triglyceride levels showed significant variation among the groups (*p* = 0.006), with non-responders exhibiting the highest mean triglyceride levels at 141.15 mg/dL, which was notably higher than those observed in partial (111.30 mg/dL) and full responders (112.36 mg/dL).

Cholesterol levels also significantly differed across the groups (*p* = 0.001), with non-responders showing the highest mean cholesterol level at 180.85 mg/dL.

Conversely, no significant differences were found in LDL (*p* = 0.658), HDL (*p* = 0.537), and prolactin levels (*p* = 0.464) across the responder categories.

A statistical analysis of genetic polymorphisms showed no significant differences among the study groups and is presented in Table 4. Extensive metabolizers comprised 25 non-responders (64.1%), 31 partial responders (58.5%) and 5 full responders (45.5%), while intermediate and poor metabolizers together were represented by 14 non-responders (35.9%), 22 partial responders (51.5%) and 6 full responders (54.5%) (*p* = 0.532). Moreover, there were no statistically significant differences between patients at first episode (*p* = 0.581) and those with chronic psychosis (*p* = 0.316) regarding genetic polymorphisms according to response to risperidone treatment.

The statistical analysis presented in Table 5 reveals significant differences in the incidence of ADR between the two groups, with a *p*-value of less than 0.001. Specifically, EM experienced fewer ADR, with 77.42% reporting no adverse effects, compared to only 27.27% of IMs and PMs. The most significant difference observed in the adverse reactions was that IMs and PMs had notably higher reports of tremor (18.18% vs. 9.68%), stiffness (31.82% vs. 6.45%), and combined tremor and stiffness (22.73% vs. 6.45%) (*p* < 0.001). This significant variance supports the hypothesis that CYP2D6 polymorphisms influence the risk of ADR, suggesting that IMs and PMs, who metabolize risperidone less efficiently, may be more susceptible to side effects.

Regarding the laboratory analysis, no significant differences were observed in blood glucose, LDL, HDL, triglycerides and prolactin levels between the two groups, indicating that these biochemical markers are not prominently influenced by CYP2D6 metabolizer status in the context of risperidone treatment in the current analysis. However, cholesterol levels did differ significantly (*p* = 0.002), with EM showing higher average levels (171.81 ± 43.19 mg/dL) compared to IM and PM (160.18 ± 32.78 mg/dL).

In terms of risperidone dosage, there was no significance (*p* = 0.052), with IM and PM being less likely to be on the highest dose range (≥5 mg/day). However, the duration of treatment did not show significant differences between the groups (*p* = 0.722).

### 2.2. Pharmacogenetic Predictions of CYP2D6 Phenotypes’ Impact on Treatment Response and Adverse Drug Reactions

The correlation between CYP2D6*4 variant and ADR was statistically significant, with a rho value of 0.23 and a *p*-value of 0.021, suggesting a moderate association where genetic variations might predict the frequency or severity of ADR. This finding supports the hypothesis that some variants of CYP2D6 can influence how patients respond to risperidone, particularly in terms of experiencing side effects, which is crucial for tailoring personalized treatment plans. The results are presented in Table 6.

The correlation between CYP2D6*4 variant and dosage of risperidone was not significant (rho = −0.18, *p* = 0.067), indicating that while there was a slight negative association, genetic variation did not significantly dictate the dosage required.

The duration of treatment showed a very weak and non-significant correlation with CYP2D6*4 variant (rho = 0.02, *p* = 0.807), indicating that genetic variations did not influence how long patients were treated. However, the duration did show a significant correlation with ADR (rho = 0.21, *p* = 0.030), suggesting that patients experiencing more severe or frequent adverse effects tended to have longer treatment durations.

There was no significant correlation between the acute or chronic phase of the disorder and CYP2D6*4 variant (rho = 0.07, *p* = 0.467), ADR (rho = −0.07, *p* = 0.456), or dosage (rho = 0.03, *p* = 0.800). However, there was a negative correlation of borderline significance between the phase of the disorder and duration of treatment (rho = −0.17, *p* = 0.081).

The analysis presented in Table 7 reveals statistically significant differences in the distribution of these genotypes among patients who experienced ADR and those who did not (chi-squared statistic: 8.47, *p*-value: 0.0145). Specifically, a higher percentage of intermediate metabolizers (52.2%) was observed among patients who experienced ADR compared to those who did not (24.2%). In contrast, extensive metabolizers (45.7%) constituted a smaller proportion among those with ADR versus those without (68.9%). This distribution suggests that individuals with the IM genotype may be more susceptible to experiencing ADR when treated with risperidone, affirming part of this study’s hypothesis concerning the influence of CYP2D6 polymorphisms on drug response.

Table 8 details a logistic regression analysis that quantified the risk of ADR based on CYP2D6 metabolizer type. The regression coefficients provide estimates of the impact of being an EM, IM or PM on the likelihood of experiencing ADR, controlling for other factors. The intercept was significant (*p* = 0.011), indicating a baseline propensity towards ADR occurrences across the sample. Notably, the coefficients for PM (9.204, *p* = 0.002) indicate a significantly higher risk of ADR for poor metabolizers, with a wide confidence interval (CI: 1.424 to 14.984), suggesting substantial variability but a consistently elevated risk. Conversely, the coefficient for EM (-0.846, *p* = 0.034) suggests a protective effect against ADR. The result for IM, although positive, was not statistically significant (*p* = 0.236), suggesting a potential increase in ADR risk that was not conclusively supported by the data.

## 3. Discussion

Our study hypothesizes a connection between the genetic polymorphism of CYP2D6 and treatment response to risperidone, posing the question if subjects with IM and PM phenotypes would show a lesser reduction in PANSS total score than EMs. To our knowledge, this is the first study on Romanian adult schizophrenia spectrum disorders’ patients treated with risperidone monotherapy. Despite the fact that no significant difference could be established between treatment response categories and genetic polymorphisms, our results are similar to those of others [48]. A study performed on 76 patients diagnosed with schizophrenia showed significant differences in treatment response to risperidone measured by PANSS in subjects with CYP2D6*10 [37]. On the other hand, in a first episode psychosis cohort of 83 subjects genotyped for *3, *4 and *6 alleles, a correlation between PM and lower hydroxy-risperidone plasma levels was observed, but with no impact on the treatment response [48]. This may be due to a reduced number of poor metabolizers encountered, as in our study. However, the lack of clinical outcome regarding treatment response to risperidone according to the metabolizing phenotypes in this study is sustained by similar results [49] and contradicted by several studies where the genetic polymorphisms showed a significant impact on the clinical outcome [34,50,51]. Although, several studies have shown significant differences in the CYP2D6 metabolizing phenotypes and the risperidone plasma levels, the impact on the treatment response has inconsistent results, especially in relation solely to risperidone; thus, our results may constitute a background for future studies including risperidone active moiety plasma concentration correlations.

Apart from treatment response in schizophrenia patients, the frequency and severity of adverse drug reactions are even more important for patients’ adherence to psychiatric treatment. We refer to treatment adherence by the degree to which patient’s behavior is in concordance to medical recommendations, jointly established between the patient and the physician. Factors involved in treatment adherence in schizophrenia patients are multifaceted and encompass various aspects [52,53], among which are lack of insight, psychopathology, substance use disorder, issues associated with medication, stigma, cultural and socio-economic aspects. Non-adherence to psychiatric treatment in schizophrenia patients is estimated around 50% [52], where non-adherence particularly related to adverse drug reactions may result in exacerbation of illness, reduce treatment efficacy, treatment resistance and increased costs [54]. Specifically, our results show that patients with EM phenotypes experienced fewer ADR, compared to the patients with IM and PM phenotypes. Oppositely, intermediate and poor metabolizers experienced significantly more often tremor, stiffness and both tremor and stiffness. This significant variation according to metabolization status supports the hypothesis that CYP2D6 polymorphisms influence the risk of ADR and is strengthened by studies with similar results [37,49,55]. Thus, IM and PM seem to metabolize risperidone less efficiently, making patients prone to extrapyramidal side effects. Thus, patients with CYP2D6*4 variants could be susceptible to poor treatment adherence to risperidone treatment, making them even more vulnerable for somatic complications and treatment resistance. Moreover, several studies showed higher health services costs related to poor treatment adherence [53,56]. However, the result for subjects with the intermediate metabolizer phenotype did not reach statistical significance, though positive and sustained by other studies with significant results [21,48]. Therefore, the risk of ADR related to genetic variations of CYP2D6*4 variant in the current study, which could help predict the frequency or severity of extrapyramidal side effects, is an important aspect in order to establishing personalized approaches and raise treatment adherence.

Interestingly, our study revealed significant differences regarding treatment duration among groups. Multiple factors have been previously found to account for the treatment response to risperidone in psychosis, with genetic polymorphisms, metabolic factors and the episode of illness duration being the most studied [24,57,58]. Likewise, the interrelation between them could explain the differences in the treatment duration in the current study as well. Significant differences were observed in the duration of treatment when comparing first episode psychotic subjects with chronic patients, those from the last category being more likely to have longer treatment durations unrelated to the genotype, reflecting both the severity of disease and the possible establishment of treatment resistance.

A laboratory analysis showed significant differences in certain biochemical parameters when comparing patients at first episode and chronic ones, but these were not related to the genotypes. Blood glucose levels were significantly lower in chronic patients compared to acute patients, which could indicate metabolic changes over the course of the illness. A study performed on autism spectrum disorder patients treated with risperidone revealed no significant impact on blood glucose levels in association with CYP2D6 genotypes, but instead significant changes correlated with other polymorphisms [59] showing more complex mechanisms involved in the metabolic pathways. On the other hand, the measurement of blood glucose level does not appreciate the overall level of blood sugar over a longer of period and could be imprecise. However, it can reflect an acute response, specific to drug-naïve patients, which is supported by experimental results on murine models treated with risperidone [60]. Moreover, our laboratory tests were performed after two weeks of the risperidone treatment period and the blood glucose levels could suffer changes later over the course of treatment. While a study measuring blood glucose levels after 8 and 12 weeks of treatment found significant increases in patients treated with risperidone, olanzapine and quetiapine [61], another study found no significant differences after 10 weeks of risperidone treatment in a sample of 125 patients treated with risperidone monotherapy [62].

Additionally, LDL cholesterol levels were significantly higher in chronic patients of the current study (*p* = 0.012), suggesting potential long-term shifts in lipid metabolism due to prolonged illness or treatment. Although, this result did not correlate with the genetic variants, it is extremely important, suggesting that risperidone antipsychotic medication alone increases the risk for metabolic changes, which could be augmented in patients with IM and PM phenotypes [46,63].

Prolactin also showed significant differences (*p* = 0.028), with acute patients having higher levels, which could be associated with more intense risperidone treatment or acute response mechanisms. Although in the present study, prolactin levels were not correlated with the study phenotypes, several studies found significant association between the elevation of prolactin and certain genotypes such as CYP2D6*2, *10, *65 [37] in adults and CYP2D6*3, *4, *41 in pediatric population [44]. Another possible explanation considering data in the literature could be that drug-naïve patients at first episode psychosis seem to have increased prolactin levels unrelated to antipsychotic treatment, suggesting that hyperprolactinemia may be a relevant characteristic of schizophrenia spectrum disorder, regardless of antipsychotic medication [64]. Therefore, people at first episode of psychosis deficient of null alleles of CYP2D6 phenotypes could be at increased risk for hyperprolactinemia, amenorrhea and galactorrhea, accordingly.

The present study acknowledges several limitations. Firstly, following treatment response grouping, the number of subjects was reduced considerably. Moreover, patients with two null alleles clinically represented as poor metabolizers consisted of only four patients, making it impossible to separate IM from PM in the statistical analyses. Secondly, there was only one variant of CYP2D6 analyzed, although it is the most frequent encountered in the Caucasian population impacting antipsychotic response, several other types that could be equally important should be evaluated. Thirdly, patients were evaluated with PANSS only on admission and at discharge, with different lengths of hospitalization among subjects. Additionally, the plasma concentration of risperidone and its active metabolite were not evaluated. Lastly, the relatively small sample of PM patients suggest a cautious interpretation of its involvement in ADR. Further replication of the study results might be needed.

Future prospective studies on larger cohorts would be advisable to clarify the complex relation between CYP2D6 variants and the plasma concentration of risperidone and its active metabolites and the risk for extrapyramidal side effects and metabolic imbalances.

## 4. Material and Methods

### 4.1. Patients and Study Design

The sample size was calculated based on the reported prevalence of schizophrenia in the general population of 1% [3], with a 99% confidence level, and a 5% margin of error, resulting in a minimum of 27 participants to reach significance. The statistical power was chosen as 80%.

We performed an observational study including patients between the age of 19 and 74 years old admitted between November 2021 and January 2024 to the Psychiatric Ward of “Pius Brinzeu” County Emergency Hospital from Timisoara, Romania for acute symptoms of SSD, according to DSM-5. With respect to eligibility for the study, patients had to meet the following criteria: (1) first psychotic episode or a diagnostic of chronic psychotic disorder diagnosed for at least 3 years, (2) treated with risperidone as antipsychotic monotherapy, (3) agreed to take part and signed an informed consent. The risperidone treatment was orally administered, and the dosages were gradually increased from a low dose to a therapeutic dose during the hospitalization, stabilized within 1 week (2–8 mg/day) according to standard clinical practice. The dosage adjustment was blinded from the genotype. The following constituted exclusion criteria: (1) patients with concomitant administration of other antipsychotic treatments or other medication that could interfere with the CYP2D6 enzymatic metabolization of risperidone, (2) subjects suffering from comorbid disorders that could interfere with the treatment outcomes (neurological disorders, renal and hepatic disorders), (3) patients with baseline impaired hepatic or renal function tests, and (4) subjects with drug-induced psychotic symptoms or those using drugs in the last month prior to admission. The socio-demographics, clinical data and laboratory samples were collected from the patients.

The PICO model for the study consisted of the following: P (population): SSD patients according to DSM-5 criteria, I (intervention): risperidone treatment, C (comparison): patients with normal enzymatic activity of CYP2D6 (EM) and patients with either very low enzymatic activity (PM) or with intermediate enzymatic activity (IM), and O (outcomes): treatment response (measure by PANSS) and adverse drug reactions.

### 4.2. Ethics Approval

Participants were informed about the study’s aims and implications and signed a written informed consent. The current study was performed conforming to the Helsinki Declaration Guidelines for scientific experiments involving human subjects and the ethical approval was received in 2021 with the approval number 10 from the Ethical Board of the “Victor Babes” University of Medicine and Pharmacy Timisoara.

### 4.3. Clinical Assessment

Patients’ response to antipsychotic treatment was assessed in a blinded fashion to the genotyping procedure by 2 researchers (M.B and L.D) using PANSS on admission and before discharge. The questionnaire comprises 30 items and 3 subscales: Positive Scale (7 items), Negative Scale (7 items) and General Psychopathology Scale (16 items). Each item is evaluated on a 7-point scale, as follows: 1-absent, 2-minimal, 3-mild, 4-moderate, 5-moderate-severe, 6-severe and, 7-extreme. The ranging scores are the following: PANSS Total score between 30 and 280 points, Positive Scale and Negative Scale scores between 7 and 49 points each, and General Scale score between 16 and 112 points. PANSS has a good internal consistency with Cronbach alpha’s between 0.70 and 0.85 [65,66].

The participants were grouped in three categories based on the percentage of PANSS scores calculated as the difference between total scores obtained at admission and total scores at discharge, as follows: non-responders with a lower reduction than 20% in PANSS total score, partial responders with a PANSS total score reduction between 20% and 40% and full responders with 40% or more reduction in PANSS total score. The PANSS threshold was established based on existing research on treatment response in patients with SSD [67,68].

Extrapyramidal side effects were evaluated 2 weeks after the risperidone treatment was started using the Simpson Angus Scale (SAS). Chronic patients undergoing other antipsychotic treatment prior to admission underwent an additional phase of a 7-day wash-out to eliminate the previous antipsychotic medication from the system before starting risperidone treatment. Subjects with scores ≥ 3 on SAS, thus confirming the presence of neuroleptic induced side effects, were further separated into three clinical categories: patients with tremor, patients with rigidity and patients having both tremor and rigidity.

### 4.4. Laboratory Tests

A baseline laboratory battery of tests was performed on admission to exclude patients with possible interference on treatment outcome, such as hepatic and renal impairments, diabetes, and hormonal imbalances. After 2 weeks of treatment, blood glucose level, cholesterol, triglycerides, LDL, HDL and prolactin were tested. The laboratory test results were interpreted according to the normal ranges provided by the Medical Analysis Laboratory of “Pius Brinzeu” County Emergency Hospital from Timisoara, Romania such as: blood glucose level = 74–106 mg/dL, cholesterol = 0–199 mg/dL, LDL = 0–99 mg/dL, HDL = 40–60 mg/dL, triglycerides = 0–150 mg/dL, prolactin in men = 4.7–17.9 ng/mL, prolactin in non-pregnant women = 2.8–29.9 ng/mL and prolactin in women in menopause = 1.8–20.3 ng/mL.

### 4.5. CYP2D6 Genotyping

An additional 2 mL volume of peripheral blood was drawn on admission for pharmacogenetic analysis. The isolation of the genomic DNA was conducted in accordance to the manufacturer’s procedures of the MagCore Nucleic Acid Extraction Kit (RBC Bioscience, New Taipei City, Taiwan). The DNA was stored at −20 degrees Celsius. An Epoch Microplate Spectrophotometer (Agilent BioTek EPOCHSN, Santa Clara, CA, USA) was used to determine the DNA concentration. For the current study, one variant (rs3892097) of the CYP2D6 gene was selected for the analysis (Assay ID: C__27102431_D0). We used TaqMan Drug Metabolism Genotyping Assays (Applied Biosystems, Foster City, CA, USA) and TaqMan Genotyping Master Mix (Applied Bio systems) according to manufacturer protocols. The purified DNA was amplified in a real-time polymerase chain reaction (PCR) on the LightCycler 480 (Roche). Gene Scanning software version 1.5.1 (Roche) was used. The genotyping process was executed in a double-blind manner by two laboratory personnel. For genotyping quality control, 5% of the samples were chosen randomly. The genotyping of these samples was repeated with a reproducibility of 100%. Subjects with two null alleles were considered poor metabolizers (PMs), those with either two deficient alleles or one gene with absent enzymatic activity and one with normal activity were considered intermediate metabolizers (IMs), and those with two normal-activity alleles were considered extensive metabolizers (EMs).

### 4.6. Statistic Analysis

Descriptive and inferential statistics were performed using the IBM SPSS software version 26.0. The normality of the data was tested with the Kolmogorov–Smirnov test. Normally distributed data were represented as mean and standard deviation, while categorical variables were represented as absolute and percentage values. Student’s *t*-test was used for comparing means for continuous Gaussian variables, while the ANOVA was used to investigate differences in means between multiple groups. Spearman’s correlation coefficient was determined for non-parametric data, while Pearson’s correlation was used for the parametric data. A logistic regression model was built to assess the risk of adverse drug reactions based on genetic polymorphisms. A Bonferroni correction was applied for multiple simultaneous comparisons. The significance threshold was considered to be α = 0.05.

## 5. Conclusions

The present study evaluated CYP2D6 polymorphisms in relation to risperidone monotherapy. The major contribution of this study is represented by the significant correlation between CYP2D6*4 variants and the risk for adverse drug reactions that could impact treatment adherence in patients treated with risperidone. We could not establish a significant correlation between CYP2D6 polymorphisms and treatment response to risperidone treatment in patients diagnosed with schizophrenia spectrum disorders. However, significant differences could be noticed regarding blood glucose levels and prolactin levels, both showing elevated levels in first episode psychosis. In contrast, elevated LDL cholesterol levels were significantly increased in chronic patients. Laboratory tests showed no significant differences in relation to the genetic phenotype. Chronic patients experienced significantly longer treatment duration, possibly due to more severe symptoms or due to the establishment of treatment resistance. Notably, extrapyramidal side effects were associated with CYP2D6 polymorphisms, particularly in PM and IM patients. Statistical significance was reached only for PM patients, thus requiring further research for confirmation.

## Figures and Tables

**Figure 1 ijms-25-06350-f001:**
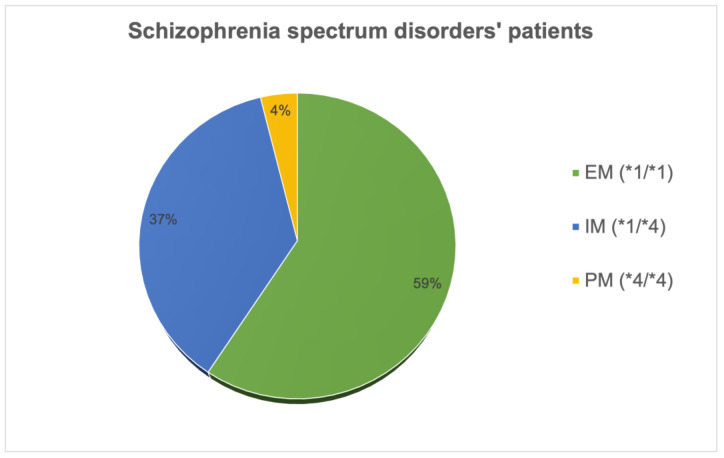
Subjects’ distribution according to phenotypes. * Schizophrenia spectrum disorder patients and genetic polymorphisms; extensive metabolizers: EM (*1/*1); intermediate metabolizers: IM (*1/*4); poor metabolizers: PM (*4/*4).

**Table 1 ijms-25-06350-t001:** Socio-demographic characteristics of the study population.

Variables	Non-Responders (n = 39)	Partial Responders (n = 53)	Full Responders (n = 11)	*p*-Value
Age (mean ± SD)	38 ± 12	41 ± 14	40 ± 11	0.14
BMI (mean ± SD)	25 ± 5	26 ± 7	23 ± 5	0.36
Age of onset (mean ± SD)	32 ± 11	33 ± 13	36 ± 11	0.64
**Age category**				0.99
18–40 years	53% (21)	49% (26)	54% (6)	
40–60 years	35% (14)	41% (22)	36% (4)	
≥60 years	10% (4)	9% (5)	9% (1)	
**Sex**				0.66
Male	56% (22)	47% (25)	54% (6)	
Female	43% (17)	52% (28)	45% (5)	
**Education**				0.7
Middle school	17% (7)	15% (8)	27% (3)	
High school	53% (21)	62% (33)	45% (5)	
University	28% (11)	22% (12)	27% (3)	
**Occupation**				0.33
Student	7% (3)	7% (4)	9% (1)	
Employed	23% (9)	35% (19)	63% (7)	
Unemployed	33% (13)	22% (12)	18% (2)	
Retired	2% (1)	0% (0)	0% (0)	
Ill-health retired	33% (13)	33% (18)	9% (1)	

Treatment response was based on PANSS threshold described in the materials and methods; SD: standard deviation; α value after Bonferroni correction = 0.007.

**Table 2 ijms-25-06350-t002:** Clinical characteristics of the study population.

Variables	Non-Responders (n = 39)	Partial Responders (n = 53)	Full Responders (n = 11)	*p*-Value
**Diagnosis**				0.3
Schizophrenia (1)	17% (7)	22% (12)	9% (1)	
Schizoaffective disorder (2)	15% (6)	16% (9)	18% (2)	
Schizophreniform disorder (3)	43% (17)	50% (27)	72% (8)	
Delusional disorder (4)	23% (9)	9% (5)	0% (0)	
**Number of episodes**				0.54
1	33% (13)	39% (21)	54% (6)	
2–3	30% (12)	18% (10)	18% (2)	
>3	35% (14)	41% (22)	27% (3)	
**ADR**				0.63
None	53% (21)	56% (30)	54% (6)	
Tremor	5% (2)	13% (7)	18% (2)	
Stiffness	30% (12)	16% (9)	18% (2)	
Tremor + Stiffness	10% (4)	13% (7)	9% (1)	
**Risperidone**				
**Dose (mg/day)**				0.79
≤2	10% (4)	13% (7)	9% (1)	
3–4	56% (22)	43% (23)	54% (6)	
≥5	33% (13)	43% (23)	36% (4)	
**Duration of hospitalization and treatment (days)**				0.01
≤14	28% (11)	13% (7)	54% (6)	
15–28	25% (10)	49% (26)	18% (2)	
29–42	33% (13)	18% (10)	9% (1)	
>42	12% (5)	18% (10)	18% (2)	

Treatment response was based on PANSS threshold described in the materials and methods; ADR: adverse drug reactions; α value after Bonferroni correction = 0.010.

**Table 3 ijms-25-06350-t003:** Laboratory markers according to treatment response.

Variables (Mean ± SD)	Non-Responders (n = 39)	Partial Responders (n = 53)	Full Responders (n = 11)	*p*-Value
Blood glucose (mg/dL)	78 ± 16	76 ± 15	74 ± 19	0.02
Cholesterol(mg/dL)	180 ± 48	166 ± 39	175 ± 31	0.01
LDL (mg/dL)	111 ± 37	102 ± 34	114 ± 30	0.66
HDL (mg/dL)	45 ± 14	47 ± 12	45 ± 16	0.54
Triglycerides (mg/dL)	141 ± 60	111 ± 48	112 ± 56	0.01
Prolactin (ng/mL)	69 ± 41	77 ± 48	105 ± 90	0.46

Treatment response was based on PANSS threshold described in the materials and methods; SD: standard deviation; mean ± SD tested with ANOVA; α value after Bonferroni correction = 0.008.

**Table 4 ijms-25-06350-t004:** Genetic polymorphisms according to treatment response.

Variables	Non-Responders (n = 39)	Partial Responders (n = 53)	Full Responders (n = 11)	*p*-Value
**Polymorphism**				0.53
EM (*1/*1)	25 (64%)	31 (59%)	5 (45%)	
IM (*1/*4) + PM (*4/*4)	14 (36%)	22 (51%)	6 (55%)	
**First episode vs. Chronic**				
EM (*1/*1)	n = 25	n = 31	n = 5	0.58
First episode of psychosis	10 (40%)	16 (52%)	3 (60%)	
Chronic psychosis	15 (60%)	15 (48%)	2 (40%)	
IM (*1/*4) + PM (*4/*4)	n = 14	n = 22	n = 6	0.31
First episode of psychosis	7 (50%)	11 (50%)	5 (83%)	
Chronic psychosis	7 (50%)	11 (50%)	1 (17%)	

Treatment response was based on PANSS threshold described in the materials and methods; EM: extensive metabolizers (wild type); IM: intermediate metabolizers; PM: poor metabolizers; α value after Bonferroni correction = 0.017.

**Table 5 ijms-25-06350-t005:** Genetic polymorphisms and their impact on adverse drug reactions, laboratory tests, dosage and treatment duration.

Variables	EM (*1/*1) (n = 61)	IM (*1/*4) + PM (*4/*4) (n = 42)	*p*-Value
**ADR**			<0.00 **
None	77% (47)	27% (11)	
Tremor	9% (6)	18% (8)	
Stiffness	6% (4)	31% (13)	
Tremor + Stiffness	6% (4)	22% (10)	
**Laboratory analysis**			
Blood glucose	77± 14	76 ± 15	0.53
Cholesterol	171 ± 43	160 ± 32	0.00
LDL	105 ± 32	97 ± 36	0.79
HDL	47 ± 12	46 ± 13	0.61
Triglycerides	113 ± 49	108 ± 48	0.87
Prolactin	75 ± 42	80 ± 56	0.18
**Dose (mg/day)**			0.05 **
≤2	11% (7)	11% (5)	
3–4	40% (25)	64% (27)	
≥5	47% (29)	23% (10)	
**Duration (days)**			0.72 **
≤14	27% (17)	19% (8)	
15–28	36% (22)	45% (19)	
29–42	19% (12)	19% (8)	
>42	16% (10)	16% (7)	

**—*p*-value was calculated using the chi-square test (the difference between proportions among polymorphism groups); Treatment response was based on PANSS threshold described in the materials and methods; ADR: adverse drug reactions; EM: extensive metabolizers (wild type); IM: intermediate metabolizers; PM: poor metabolizers; α value after Bonferroni correction = 0.012.

**Table 6 ijms-25-06350-t006:** Spearman correlation of genetic polymorphisms and adverse drug reactions.

Rho/*p*-Value	Polymorphism	ADR	Dose (mg/Day)	Duration (Days)	First Episode/Chronic Psychosis
Polymorphism	1				
ADR	0.23 (*p* = 0.02)	1			
Dose (mg/day)	−0.18 (*p* = 0.06)	0.12 (*p* = 0.21)	1		
Duration (days)	0.02 (*p* = 0.80)	0.21 (*p* = 0.03)	0.11 (*p* = 0.27)	1	
First episode psychosis/Chronic	0.07 (*p* = 0.46)	−0.07 (*p* = 0.45)	0.03 (*p* = 0.80)	−0.17 (*p* = 0.081)	1

Treatment response was based on PANSS threshold described in the materials and methods; ADR: adverse drug reactions.

**Table 7 ijms-25-06350-t007:** The distribution of genetic polymorphisms according to adverse drug reactions.

Variables	No ADR (n = 58)	ADR Present (n = 46)	*p*-Value
CYP2D6*4			0.01
EM (*1/*1)	40 (69%)	21 (46%)	
IM (*1/*4)	14 (24%)	24 (52%)	
PM (*4/*4)	4 (7%)	1 (2%)	

ADR: Adverse Drug Reactions; EM: extensive metabolizers (wild type); IM: intermediate metabolizers; PM: poor metabolizers. Chi-squared statistic: 8.47; *p*-value: 0.0145.

**Table 8 ijms-25-06350-t008:** Logistic regression results for ADR risk based on polymorphism type.

Coefficient	Estimate	SE	z-Value	*p*-Value	95% CI Lower	95% CI Upper
Intercept	−1.23	0.48	−2.53	0.01	−1.47	−0.22
EM	−0.84	0.24	−2.44	0.03	−0.16	−1.52
IM	1.65	0.30	2.16	0.23	0.06	1.25
PM	9.20	0.39	3.02	0.00	1.4	14

EM: extensive metabolizers (wild type); IM: intermediate metabolizers; PM: poor metabolizers; ADR: Adverse Drug Reactions; SE: standard error; CI: Confidence Interval.

## Data Availability

Data available on request.

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
