# Peer review of "Into a Deeper Understanding of CYP2D6’s Role in Risperidone Monotherapy and the Potential Side Effects in Schizophrenia Spectrum Disorders"

_ijms, 2024, doi:10.3390/ijms25126350_

Round 1

Reviewer 1 Report

Comments and Suggestions for Authors

This study aimed to evaluate the association between CYP2D6*4 allele and outcomes such as adverse drug reactions, number of responders, dose, and laboratory parameters in patients with schizophrenia under ripseridone therapy. Although findings that connect genetic polymorphisms to clinical outcomes are of interest, some adjustments would improve the quality of the paper and some limitations should be considered and discussed in more detail.

Some specific comments and suggestions:

Title:

-       Currently not clear based on the title that risperidone was the focus of the study

Introduction:

-       A refrence could be added after the first sentence of the introduction section

-       “…76% more common among men…”: Better to provide the information how many times higher

-       Some information should be added about the current knowledge and recommendations in the context of risperidone therapy based on genotype (as provided for example by www.pharmgkb.org) and the roles of the mother substance and the metabolite regarding therapy failures and adverse effects

-       Currently not clear in the aim/hypothesis section why you focused only on this one allele

Methods:

-       “…resulting in a minimum of 27 participants to reach significance”: Is this number by group or total? What was the rational for inlcuding 107 patients at the end?

-       Currently not mentioned/explained in the methods how you grouped the different genotypes (which alleles were grouped as EM, PM, and IM) and why

-       Section CYP2D6 genotyping: Some more information should be provided about the alleles that were tested/included

-       “Continuous data were represented as mean and Standard deviation…”: Were these data normally distributed? If not, they should be presentes as median and e.g. ranges

-       “Statistical power was chosen for 80%...at a=0.05”: Shouldn’t that be mentioned together with the information how many patients were needed to reach significance, which is provided in the beginning of the methods?

Results:

-       Currently too many results presented, most of them not focusing on the genotype, and with some different analyses investigating the same research question. I would suggest focusing on the results based on the genotype (main study question) and presenting the rest of the results in less detail. Also avoid repetition of the same findings in the main text and in tables/figures, it’s sufficient if you mention them only once

-       Figure 1 could be deleted or adjusted to be more informative, currently no numbers are shown and the information is also provided in the text

-       Currently in the results you provide a lot of interpretations (e.g. “This could suggest a metabolic adjustment…”, “This significant variance supports the hypothesis…”, “This finding could suggest a differential impact…”, etc.), these should be moved in the discussion, in the results section only the findings should be provided, not their implications and possible suggestions

-       Table 3: Which was the timepoint of the laboratory markers shown here?

-       Table 4: IM and PM are shown together as one group, you should explain somewhere before this Table why you chose to do so

-       Table 5: ADR p-value <0.001, for which ADR comparisons of all those that are currently shown?

-       Terms such as “a marginal trend” should be avoided, just state if it was statistically significant or not

-       You currently compare many different variables and conditions, did you correct for multiple comparisons? If yes, this should be clarified in the methods section

-       “The correlation between CYP2D6*4 variant and ADR…”: The genotypes are not nummerical variables, how did you perform a Spearman correlation with those?

-       You currently use many different approaches to investigate the relationship between genotype and ADR (presented among others in the Tables 7, 8, and 9), I would suggest consulting a statistician and chosing one/the most appropriate method in this case

Discussion:

-       Currently a lot of repetition of the results, the discussion should describe the data qualitatively and not repeat the numbers from the results section

-       Currently not clear, what this study adds for th eclinical practice, especially since you only investigated one allele of reduced function while many others might have an influence on the studied outcomes and some recommendations are already available in the scientific literature. It would be of advantange to discuss your findings in the context of the already available information regarding genotypes, with a focus on what your study adds and the implications for the clinical management of patients

Comments on the Quality of English Language

Minor editing required

Author Response

Thank you very much for taking the time to review this manuscript.

Reviewer 2 Report

Comments and Suggestions for Authors

The manuscript of Bondrescu et al. preents a  study on the compared effect of risperidone treatment on schizophrenic patients in function of their CYP2D6 genotype.

The study is well written. But I am not a neurologist and more a P450 specialist. I don’t understand some of the bases of this study.

It looks that 9-hydroxy-resperidone (also sold as paliperidone) is the major metabolite of active CYP2D6. It has also similar pharmacological effect as risperidon. However Risperidone seems to have more activity in causing stiffness and other extrapyramidal side effects.

CYP2D6 slow metabolizer should metabolize slowly the parent drug resperidone increasing its steady–state concentration thus causing more side effect?  They should also not produce the metabolite 9-hydroxyrisperidone.

Thus I wonder why the concentration of these two compound have not been measured as you will do in a clinical study? If the blod probe wer taken for instance at two hours  (or one hour) after drug intake and after about one week of treatment, you would have a comparable value of interest.

Another point that is not clear: Is the dose of resperidone the same for each patient or is it adapted to its weight or to is reaction to the drug or even to it 2D6 genotype (does the clinician know it at the start of the study?)

I have a few more questions and remarks I put below:

            Page 2 line 96: 9-hydroxylation.

            Page 3 top : It seem to me that you should explain the change in circulating concentration of resperidone that increase in low and intermediate metabolizers, thus explaining adverse reaction by excess drug (in this case).

            Concerning the tables ; I am always puzzled why scientists always put so many digits in tables when none of the decimal digit is significant. These digits are given by your computer but mean nothing. Thus they are not needed. tIf you suppress them this would make the table lighter and more comprehensible. Give only two digits, also for the p value.

            Page 8 : Not being a psychatrist , I don't understand the duration variation of the treatment. Is it until one reaches a certain response to some test.

If I understand you do a SAS test before starting the treatment and one after 2 weeks of treatment and then separate patients having a high SAS score? As a index of drug toxicity (or excess)

Thus I wonder why you did not monitor the level of risperinone and metabolite in the plasma of patients. Obviously this should be done with a similar timing after drug intake: For intance at 1 h or 2 hours .

Slow metaboliser (PM) would probably have a high concentration of Risperidone in blood.

            Page 10 : do you mean that the dose of risperidone was adapted by the clinician? Then on which criterium? Did the clinician know the CYP2D6 genotype of the patient?

            Page 12 : This figure 2 is contradictory with the abstract where you state that the ADR were more present among IM and PM. This is true for IM not for PM?

As a whole the study reports results of a cohort of schizophrenic patients treated with risperidone.

But to me it the message that the authors like to bring is not clear. Perhaps the above questions and remarks may help them reformulate their finding so that the message is clear.

I think that the study still need some good polishing.

Some statement in the interpretation of the results seems to me contradictory. May be I don’t understand them well.

Comments on the Quality of English Language

There are only a few spelling errors. Otherwise good quality english.

Author Response

(The authors gave the same response as above.)

Reviewer 3 Report

Comments and Suggestions for Authors

In this study, Bondrescu and colleagues divided a sample of 103 people with schizophrenia spectrum disorders treated with risperidone monotherapy into three groups of non-responders, partial responders, and full responders, exploring metabolic correlates, side effects, and CYP2D6 genotype.

This study is very comprehensive and interesting, and may represent a relevant contribution to the scientific literature about antipsychotic treatment and its effectiveness, side effects, and metabolic implications.

I have no critical concerns about how the study was conducted and reported. Yet, I have some observations that can hopefully help the Authors further increase the quality of their article:

1.     The title is too vague. It should be more focused on the study, also indicating the study design with a commonly used term as suggested by the STROBE statement for the reporting of observational research (https://www.strobe-statement.org/checklists/). To me, it should sound somewhat like “The relationship between CYP2D6, response to treatment, and side effects in people with Schizophrenia Spectrum Disorders taking risperidone monotherapy”.

2.     Abstract, L 32: “correlated” is misspelled.

3.     The Introduction section is overly long. I suggest that the Authors synthesize it, especially the first part (until L 63).

4.     Introduction: the risk of hyperprolactinemia induced by risperidone, very relevant to this drug [https://doi.org/10.1016/j.psychres.2016.04.001] and to this study, is not mentioned.

5.     The number of subjects ultimately included in the study (L 117; LL 132-136) should be reported in the Results section (around LL 215-217), following the STROBE statement. The Methods section should only report inclusion and exclusion criteria + sample size calculation (correctly done).

6.     At LL 292-296, do the Authors refer to length of hospitalization or to duration of risperidone treatment (meaning that the drug was tapered off immediately after)? This should be made clearer.

7.     Regarding prolactin, the Authors report that participants with first-episode psychosis showed significantly higher prolactin levels (p=0.034), suggesting that this could be associated with more intense risperidone treatment or acute response mechanisms and that people at first episode of psychosis with deficient of null alleles of CYP2D6 phenotypes could be at increased risk for hyperprolactinemia, amenorrhea and galactorrhea. However, the Authors could provide a further layer to their interpretation by taking into account that solid evidence demonstrates that drug-naïve people with first-episode psychosis have higher blood prolactin levels, which may mean that prolactin increase may be relevant to schizophrenia spectrum disorders regardless of antipsychotic treatment [https://doi.org/10.1016/j.psyneuen.2023.106392].

Author Response

(The authors gave the same response as above.)

Round 2

Reviewer 1 Report

Comments and Suggestions for Authors

Thank you for addressing the previous comments. Some suggestions for some issues that might still need your attention:

- There is still a lot of repetition between the text and the Tables in the results, e.g.: “Following statistical analysis on levels of education, among subjects with middle school studies, 7 persons were non-responders (17.94%), 8 persons (15.09%) were partial responders, and 3 persons (27.27%) were full responders. 21 of the non-responsive subjects (53.84), 33 of the partial responders (62.26%) and 5 full responders (45.45%) completed high school studies. Finally, 11 non-responders (28.20%), 12 partial responders (22.64%) and only 3 full responders (27.27%) graduated university. There was no statistical difference among the three groups (p=0.707).” -> Those results are already presented in Table 1, no need to repeat also in such detail in the main text. Similar comment for other parts of the results section. 

- You should check your reported significant differences again now that you have corrected for multiple compariosns, e.g. “However, the duration of hospitalization did show significant differences among the 675 groups (p=0.017)”, but based on the footnote of Table 2 significance would be 0.010 after Bonferroni correction. You might also have to adjust some parts of the discussion accordingly.

- Table 3: "Triglycerides p = 0.00"; do you mean < 0.001?

- Table 5: You have now added that "the p-value for ADR was calculated with chi-square (the difference between proportions of adverse reactions between polymorphisms)" but it is still not clear for which of the ADRs shown the difference was significant

- “The correlation between CYP2D6*4 variant and ADR…”: You have now added that Spearman’s correlation was performed for non-parametric and Pearson’s for the parametric data, but you still didn’t answer the question how you performed these correlations with non nummerical data, i..e the genotypes.

- To the comment that you currently use many different approaches to investigate the relationship between genotype and ADR (presented among others in the Tables 7, 8, and 9), you answered that you did so because there are still many unknowns of the CYP2D6 polymorphisms and you attempted to determine all possible evidence of associations. However, using many different statistical approaches to investigate the same question might not be the best way to do that. I would still suggest consulting a statistician regarding this issue.

Comments on the Quality of English Language

Minor editing required.

Author Response

Thank you again very much for taking the time to review this manuscript. Please find the detailed responses below and the corresponding revisions in track changes in the re-submitted manuscript.
